# Cyclin B Export to the Cytoplasm via the Nup62 Subcomplex and Subsequent Rapid Nuclear Import Are Required for the Initiation of *Drosophila* Male Meiosis

**DOI:** 10.3390/cells12222611

**Published:** 2023-11-11

**Authors:** Kanta Yamazoe, Yoshihiro H. Inoue

**Affiliations:** Biomedical Research Center, Graduate School of Science and Technology, Kyoto Institute of Technology, Matsugasaki, Sakyo, Kyoto 606-0962, Japan; d1841005@edu.kit.ac.jp

**Keywords:** *Drosophila*, male meiosis, cyclin B, cyclin-dependent kinase 1, cyclin-dependent kinase inhibitor

## Abstract

The cyclin-dependent kinase 1 (Cdk1)–cyclin B (CycB) complex plays critical roles in cell-cycle regulation. Before *Drosophila* male meiosis, CycB is exported from the nucleus to the cytoplasm via the nuclear porin 62kD (Nup62) subcomplex of the nuclear pore complex. When this export is inhibited, Cdk1 is not activated, and meiosis does not initiate. We investigated the mechanism that controls the cellular localization and activation of Cdk1. Cdk1–CycB continuously shuttled into and out of the nucleus before meiosis. Overexpression of CycB, but not that of CycB with nuclear localization signal sequences, rescued reduced cytoplasmic CycB and inhibition of meiosis in *Nup62*-silenced cells. Full-scale Cdk1 activation occurred in the nucleus shortly after its rapid nuclear entry. Cdk1-dependent centrosome separation did not occur in *Nup62-*silenced cells, whereas Cdk1 interacted with Cdk-activating kinase and Twine/Cdc25C in the nuclei of *Nup62-*silenced cells, suggesting the involvement of another suppression mechanism. Silencing of *roughex* rescued Cdk1 inhibition and initiated meiosis. Nuclear export of Cdk1 ensured its escape from inhibition by a cyclin-dependent kinase inhibitor. The complex re-entered the nucleus via importin β at the onset of meiosis. We propose a model regarding the dynamics and activation mechanism of Cdk1–CycB to initiate male meiosis.

## 1. Introduction

A conserved molecular mechanism that controls the initiation of cell division in eukaryotes involves the activation of cyclin-dependent kinase 1 (Cdk1), which serves as a master regulator of the M phase of mitosis and meiosis [1,2]. In eukaryotes, the following three conditions are indispensable for activating this protein kinase: complex formation with its regulatory subunit, cyclin B (CycB); phosphorylation of Thr^161^ of Cdk1; and removal of phosphate groups from Thr^14^ and Tyr^15^, both of which are involved in the negative regulation of the kinase phosphorylated by Wee1/Myt1 [3,4,5]. Cdk1 is activated at the onset of the M phase via dephosphorylation of Thr^14^ and Tyr^15^ by cell division cycle 25 (Cdc25) orthologues. Thr^161^ of Cdk1 also needs to be phosphorylated by Cdk-activating kinase (CAK) [6]. In addition to Cdk1 modification, another type of inhibitors known as Cdk inhibitors (CKIs), such as p21, play an important role in controlling the cell cycle. CKIs were originally identified as negative factors that bind to suppress Cdk activity at the G1/S phase and also affect CycB-Cdk1 during the G2/M transition [7,8]. These inhibitors need to be released from Cdk1 before the onset of the M phase [9,10]. From later stages of the G2 phase towards the beginning of the M phase, Cdk1 activity depends on an increase in the expression of CycB. In vitro assays using animal oocyte extracts have revealed that Cdks are activated progressively [11,12,13]. A small population of the CycB–Cdk1 complex is first activated by a trigger [14]. Consequently, the balance between Cdc25 and Wee1/Myt1 activities is shifted so that Cdc25 activity becomes predominant. CycB–Cdk1 further accelerates dephosphorylation of the kinase via positive feedback loops, leading to maximal activation [13,15,16]. In contrast, a double-negative feedback loop implemented by the inactivation of a counteracting phosphatase by Cdk1 can also contribute to Cdk1’s own activation [17]. In addition, the subcellular localization of Cdk1 and its regulatory factors and the timing of their migration to other compartments are considered critical points for mitotic entry in mammalian cells [18,19,20,21]. In the G2 phase, CYCB1 is enriched in the cytoplasm but continuously shuttles into and out of the nucleus until shortly before the onset of mitosis [21]. Mitosis is triggered by the activation of Cdk1–CycB and its translocation from the cytoplasm to the nucleus. The spatial and feedback regulation ensures a rapid and irreversible transition from interphase to mitosis.

Much progress has already been made in elucidating special regulatory activities that control Cdk1 activation during the G2/M transition in mitosis. However, several issues remain to be uncovered regarding the mechanism of meiotic initiation. Meiosis is expected to be highly susceptible to spatial and temporal control of the cell cycle in cooperation with the developmental program. For example, in mouse oocytes, spatial regulation of anaphase-promoting complex (APC)/C^Cdh1^-induced CycB degradation maintains G2 arrest of oocytes for several years [22]. The stepwise activation of Cdk1 may, rather, play a more important role in meiosis than in the mitotic cell cycle. In *Drosophila*, the developmental program and cell-cycle progression in meiosis have been better studied [23]. As the meiotic cycle in *Drosophila* generally constitutes a prolonged G2-like growth period, the timing of meiosis initiation is expected to be strictly regulated. However, the mechanism by which Cdk1-dependent phosphorylation is timed to occur shortly before the nuclear envelope breaks down remains to be explored. With reference to the regulatory mechanisms of mitotic initiation, we can expect a similar regulatory mechanism to initiate male meiosis in *Drosophila* [24]. In contrast, several specific regulations separate from the core regulatory system take place during meiosis. For example, a Cdc25 orthologue encoded by *twine* plays a meiosis-specific role in activating Cdk1 before the onset of meiosis during oogenesis and spermatogenesis, whereas *string* is required at the initiation of mitotic events during embryogenesis and those during the development of germline stem cells and their progenitor cells [25,26,27]. *cycB* mRNA is expressed at low levels in the spermatogonia during mitotic proliferation. Then, it is downregulated after the completion of mitotic divisions and re-expressed at high levels in spermatocytes during the growth phase before meiosis [28]. In contrast, CycB protein levels in spermatocytes remain low until spermatocytes enter the G2/M transition after the appearance of mRNA. *CycB* translation is repressed until before the onset of male meiosis [29] by two proteins that bind to *cycB* mRNA in spermatocytes. CycB accumulates in the cytoplasm prior to the initiation of chromatin condensation, remains at a high level during prophase, and then enters the nucleus at the onset of meiosis.

*Drosophila* spermatocytes before or during meiosis I offer several advantages with respect to the investigation of cell-cycle regulation at the G2/M phase. Identifying and observing meiotic cells is easy due to the large cell size, which originates the remarkable cell growth. This facilitates the observation of the subcellular localization of specific regulatory proteins [23]. Nevertheless, comparing the spermatocytes at similar developmental stages in different cysts is not easy. The growth phase has been classified into six stages, S1–6, based on the chromatin morphology and the intracellular structure of pre-meiotic spermatocytes [30]. Recently, the characteristic size and morphology of the nucleolus in the growth phase have allowed us to precisely identify the developmental stages of spermatocytes [31].

The small pores that penetrate the nuclear membrane are called nuclear pore complexes (NPCs) and play a critical role in regulating the nuclear–cytoplasmic transport of mRNAs and proteins. [32]. The NPCs are constructed of more than 30 types of nucleoporins (Nups), and these are highly conserved among eukaryotes. The protein transport between the cytoplasm and the nucleus through the NPCs requires a group of proteins called Karyopherins [33]. These group proteins are classified into importins, which help proteins get into the nucleus by binding to nuclear localization sequences (NLSs), and exportins, which help proteins get out of the nucleus. Members of the importin-β family bind cargo proteins to transport them while mediating interactions with the NPCs [34]. Surprisingly, no spermatocytes undergoing meiosis have been observed in testes featuring the spermatocyte-specific depletion of components of the nuclear porin 62kD (Nup62) subcomplex, which comprises the central channel of the NPC, although meiosis initiates normally in testes featuring the depletion of other NPC subcomplexes [35]. Moreover, previous research has shown that silencing *Nup62* using RNA interference (RNAi) results in the accumulation of CycB in the nucleus during the growth phase, corresponding to a prolonged G2 phase before the initiation of meiosis. This inhibits Cdk1 activation, leading to cell-cycle arrest before male meiosis [36]. However, these results are inconsistent with previous results suggesting that the precocious accumulation of CycB in the nucleus by export-defective CycB expression does not influence mitotic entry in mammalian cells [19]. This unexpected finding highlights the importance of CycB subcellular localization in cell-cycle progression before male meiosis and suggests that selective nuclear–cytoplasmic transport of cell-cycle regulators may be critical for determining the timing of meiotic initiation [36]. A constitutively active Cdk1 mutant (Cdk1^T14A Y15F^) has failed to rescue the meiotic phenotype caused by *Nup62* silencing, suggesting that the removal of inhibitory Cdk1 phosphorylation was not involved in the absence of male meiosis [33]. However, the mechanism by which meiotic initiation is hampered upon inhibition of CycB nuclear export remains unclear.

In this study, we aimed to clarify the importance of protein transport in determining when meiosis is initiated in *Drosophila* males. We performed a time-lapse observation of living pre-meiotic spermatocytes to investigate whether CycB continuously shuttles between the nucleus and cytoplasm before meiosis. Furthermore, we investigated the subcellular localization of positive regulators and the formation of possible complexes between the regulatory proteins and Cdk1. We also examined whether negative regulators were involved in inhibiting the nuclear export of Cdk1 and its activation by RNAi. Additionally, we examined whether importin β is required for the re-entry of Cdk1 into the nucleus. Based on the results of those investigations, we proposed a new model regarding the intracellular dynamics and stepwise activation of Cdk1–CycB to initiate male meiosis in *Drosophila*.

## 2. Materials and Methods

### 2.1. Drosophila Stocks

We used the following *UAS-RNAi* stocks for RNAi experiments: *P{KK108318}VIE-260B* (VDRC#100588) from the Vienna Drosophila Resource Center (VDRC; Vienna BioCenter, Vienna, Austria) was used for the depletion of *Nup62* [36], *P{TRiP.GL00262}* (BDSC#35350) for the depletion of *Cdk1* [37], and *P{TRiP.HMS00467*} (BDSC#32467) for the depletion of *rux* [38]. *P{TRiP.GL01273}* (BDSC#41845) was used for the depletion of *Fs(2)Ketel* [39]. These RNAi stocks were obtained from the Bloomington Drosophila Stock Center (BDSC; Indiana University Bloomington, Bloomington, IN, USA). To visualize CycB in living spermatocytes, *P{Ubi-p63E-CycB.GFP}* was used (a gift from J. Raff, University of Oxford, Oxford, UK) [40]. We used *M{UAS-CycB.ORF.3xHA}* (#F001154, Fly-ORF; University of Zurich, Zurich, Switzerland) for the overexpression of normal CycB [41]. We also used *P{twine-EGFP}* (a gift from J. Großhans; The Philipps University of Marburg, Marburg, Germany) [42] to express green fluorescence protein (GFP)-tagged Twine. For the induction of various dsRNAs against endogenous mRNAs in spermatocytes, we used *P{UAS-dcr2}; P{bam-GAL4::VP16}* (abbreviated as *bam-Gal4*) [35]. As controls, F1 progenies from a genetic cross between *bam-Gal4* and *w* were used and denoted as *bam>+*. *P{Sa-GFP}* was used as a marker to determine the growth phase of spermatocytes [31]. All *Drosophila melanogaster* stocks were maintained on standard cornmeal food at 25 °C, as previously described [43]. For the efficient induction of GAL4-dependent cDNA and dsRNA expression, progenies carrying the GAL4 driver and UAS transgenes were raised at 28 °C.

### 2.2. Transformation

To establish transgenic lines expressing nucleus-localized CycB, *pUAS-CycB.HA-NLS* plasmid DNA was provided by F. Sprenger (Universität Regensburg, Regensburg, Germany) [44]. Purified plasmid DNA was injected into *Drosophila* embryos via PhiC31 integrase-mediated germ-line transformation using the *Drosophila* Embryo Injection Services of BestGene, Inc. (Chino Hills, CA, USA).

### 2.3. Preparation of Post-Meiotic Spermatids

For the assessment of meiotic defects, we used a previously described protocol [36]. A pair of testes collected from pharate adult or newly eclosed adult flies (0–2 d old) were dissected in testis buffer (183 mM KCl, 47 mM NaCl, 10 mM EDTA, pH 6.8) and covered with an 18 × 18 mm coverslip (Matsunami, Osaka, Japan) to flatten the cysts. To observe fixed post-meiotic spermatids, we removed the coverslips after freezing the slides and exposed the slides to 100% methanol for 5 min at −30 °C. Then, the samples were rehydrated in 1× phosphate-buffered saline (PBS, 137.0 mM NaCl, 2.7 mM KCl, 10.1 mM Na_2_HPO_4_·12H_2_O, 1.8 mM KH_2_PO_4_). Observation was performed with a phase-contrast fluorescent microscope (IX81, Olympus, Tokyo, Japan), and images were captured with a Charge Coupled Devices (CCD) camera (Hamamatsu Photonics, Shizuoka, Japan). Image acquisition was controlled using the Metamorph (Molecular Devices, Sunnyvale, CA, USA) software.

### 2.4. Immunofluorescence

Testis cells were collected according to the protocol described in 2.3. The cells were fixed with 100% ethanol for 10 min and with 3.7% formaldehyde for 7 min. The slides were permeabilized in PBST (PBS containing 0.01% Triton-X) for 10 min and blocked with 10% normal goat serum (Wako Chemical, Osaka, Japan) in PBS. The following primary antibodies were used at the dilutions described: rabbit anti-CycB antibody (a gift from D. Glover, Cambridge University, Cambridge, UK) [45], 1/400; mouse anti-HA (6E2, Cell Signaling Technology, Danvers, MA, USA), 1/200; mouse anti-mitotic protein monoclonal 2 (MPM2) antibody (05-368, Sigma-Aldrich, St. Louis, MO, USA), 1/400; guinea pig anti-Asl (a gift from J. Raff, University of Oxford, Oxford, UK) [46], 1/800; mouse anti-Cdk7 antibody (20H5, Developmental Studies Hybridoma Bank, Iowa City, IA, USA), 1/200; rat anti-Twine antibody (a gift from E. Wieschaus, Princeton University, Princeton, NJ, USA) [47], 1/100; rabbit anti-Cdk1 antibody (06-923, Sigma-Aldrich, Burlington, MA, USA), 1/200; and mouse anti-GFP antibody (3E6, Invitrogen, Carlsbad, CA, USA), 1/200. After incubation with the primary antibodies overnight at 4 °C, the samples were washed in PBS and subsequently incubated with goat anti-mouse, rabbit, Guinea pig, or rat IgG (H+L) conjugated with Alexa Fluor 488, 555, or 647 (Invitrogen) for 2 h at 25 °C. After washing in PBS, the samples were mounted with VECTASHIELD Antifade Mounting Medium with 4′,6-diamidino-2-phenylindole (Vector Laboratories, Newark, CA, USA). Image acquisition was performed as described in Section 2.3.

### 2.5. Live-Cell Imaging of Primary Spermatocytes

To observe the nuclear–cytoplasmic transport of CycB–GFP in living primary spermatocytes, we performed a time-lapse observation as previously described [48]. A pair of testes collected from pharate adults was dissected in BRB80 buffer (80 mM PIPES, 1 mM MgCl_2_, 1 mM EGTA, pH 6.8) under mineral oil (Trinity Biotech, Bray, Ireland) on a clean coverslip. To facilitate the observation of CycB–GFP nuclear import, we inhibited exportin with leptomycin B (LMB; Cayman Chemical, Ann Arbor, MI, USA). LMB was added to BRB80 buffer to a final concentration of 10 μM. GFP fluorescence images were captured by a CCD camera (Hamamatsu Photonics, Shizuoka, Japan) at each 60 s interval. Image acquisition was controlled using the Metamorph (Molecular Devices, Sunnyvale, CA, USA) software.

### 2.6. In Situ Proximity Ligation Assay (PLA)

We performed a PLA to detect close interactions between Cdk1 and its positive regulators using a Duolink kit (Sigma-Aldrich) as previously described [36]. We used the following combinations of antibodies: mouse anti-Cdk7 and rabbit anti-Cdk1 antibodies to detect the complex containing Cdk7 and Cdk1; mouse anti-GFP and rabbit anti-Cdk1 antibodies to detect the complex containing Twine-GFP and Cdk1. Image acquisition was performed as described above. Negative control experiments were also performed to confirm that either one of the antibodies used in the set produced few PLA-positive foci.

### 2.7. Statistical Analysis

For the measurement of the fluorescence intensity, more than 20 pairs of testes were used for each genotype. The results are presented as bar graphs or line charts using Prism (Version 9, GraphPad Software, San Diego, CA, USA). The number of immunostaining foci or the area size in the pixels was calculated. Each dataset was assessed using Welch’s *t*-test, an analysis of variance (ANOVA). We used the Mann–Whitney test to compare the two groups. One-way ANOVA followed by Bonferroni’s post hoc comparison test was applied to analyze the differences in more than two groups. To show the confidence levels, the 95% confidence intervals (CIs) were determined and presented as means ± 95% CIs. Statistical significance is described in each figure legend as follows: * *p* < 0.05, ** *p* < 0.01, and *** *p* < 0.001. A *p*-value of 0.05 or less was considered statistically significant.

## 3. Results

### 3.1. CycB Exhibits Continuous Nuclear Import and Export via the Nup62 Subcomplex during the Growth Phase of Drosophila Spermatocytes before Meiosis

If the Nup62 subcomplex or a *Drosophila* exportin orthologue is depleted, CycB remains to be accumulated in the nucleus through the growth phase of spermatocytes before meiosis; in turn, Cdk1 is not activated, and meiosis does not initiate [36]. In this study, we first performed immunostaining on spermatocytes and observed the cellular localization of CycB during stages S3–S6 of the growth phase and prophase I (ProI) (Figure 1a’–e’). We determined the stages of spermatocytes using spermatocyte arrest (Sa)-GFP [31] (Figure 1a”–e”). To investigate the cellular localization of CycB, we quantified CycB fluorescence in the nucleus and cytoplasm. During S3–S5, we observed a weak fluorescent signal in the nucleus and cytoplasm. Thereafter, cytoplasmic localization became clear, and an intense signal was observed in the cytoplasm at stage S6. After Sa-GFP fluorescence indicated the disappearance of the nucleolus, CycB entered the nucleus (Figure 1e’). We quantified the mean fluorescence intensity in spermatocytes in each developmental stage (Figure 1f). The CycB fluorescence intensity in cells increased from the S3 to the S5 stage and subsequently increased sharply from the S5 to the S6 stage, just before the onset of meiosis, although the ratio of nuclear to cytoplasmic (N/C) fluorescence was almost constant, indicating a balance between nuclear entry and export. The CycB fluorescence intensity continued to increase from S6 to ProI. To verify CycB dynamics, we next performed a time-lapse observation of GFP-tagged CycB in living spermatocytes (Figure 1g–i). Under drug-free ex vivo culture conditions, CycB in pre-meiotic spermatocytes was mostly localized in the cytoplasm and nucleolus (Figure 1g). However, in the presence of LMB, which inhibited CRM1/exportin1 specifically, CycB gradually accumulated in the nucleus, although CycB–GFP fluorescence in the nucleolus did not change (Figure 1h). The GFP fluorescence intensities in the nucleus and cytoplasm became almost equal after 70 min. Inhibition of nuclear export by LMB highlighted the continuous nuclear import of CycB before meiosis. Therefore, a previous result that CycB is accumulated in the nuclei of *Nup62-*silenced spermatocytes was considered a consequence of the disruption of nuclear export [36] (Appendix A).

### 3.2. Overexpression of Normal CycB, but Not CycB Harboring a Nuclear Localization Signal (NLS), Rescued the Inhibition of Meiotic Initiation in Nup62-Depleted Spermatocytes

We addressed the mechanism by which the inhibition of CycB nuclear export via an exportin orthologue and Nup62 inhibited Cdk1 activation. We analyzed whether the persistent accumulation of CycB, which could not be exported due to *Nup62* silencing, in the nucleus was responsible for the failure of meiotic initiation. CycB carrying NLSs (*bam>CycB-NLS*) or normal CycB (*bam>CycB*) was induced in spermatocytes (Appendix A). CycB abundance was lower in the nuclei than in the cytoplasm of the control cells (*bam>+*). Ectopic expression of normal CycB in control cells did not significantly change CycB localization, whereas that of NLS-CycB increased the N/C ratio by 0.4 (Figure 2a). This indicates that NLS-CycB was accumulated in the nucleus. In contrast, the N/C ratio significantly increased as a consequence of abnormal nuclear accumulation of CycB in *Nup62-*silenced cells. Cytoplasmic CycB increased by ectopic expression of normal CycB in *Nup62-*silenced cells (*bam>CycB*, *Nup62RNAi*). In contrast, ectopic expression of NLS-CycB had no impact on the N/C ratio in *Nup62-*silenced cells (Figure 2a).

To investigate whether this change in the N/C ratio of CycB affected the initiation of meiosis, we observed the cysts of spermatids at the onion stage immediately after the completion of meiosis II. During this phase, 16 pre-meiotic spermatocytes comprising a cyst undergo two consecutive meiotic divisions synchronously. Eventually, a cyst consisting of 64 spermatids is generated after the second meiotic division. In testes with *Nup62-*silenced spermatocytes (*bam>Nup62RNAi*), all spermatid cysts consisted of 16 cells (*n* = 24), whereas every spermatid cyst contained 64 post-meiotic cells in the control testes (*n* = 39) (Figure 2b). Every cell in the abnormal cysts possessed an abnormally large nucleus. This phenotype of spermatids derived from the *Nup62-*silenced spermatocytes indicated that the spermatid cysts consisting of 16 cells failed to complete two meiotic divisions. All spermatid cysts comprised 64 cells in *bam>CycB* (*n* = 15) and *bam>CycB-NLS* (*n* = 18) testes. Interestingly, the overexpression of CycB in *Nup62-*silenced cells resulted in the generation of post-meiotic cysts consisting of 32 cells (12.5% of total spermatid cysts examined, 4 out of 32) and 64 cells (56.2% of the spermatid cysts, 18 out of 32), whereas ectopic overexpression of NLS-CycB produced cysts containing 16 spermatids only (*n* = 24) (Figure 2b). These genetic data showing that inhibition of meiotic initiation in Nup62-silenced spermatocytes was rescued by overexpression of CycB, but not NLS-CycB, suggested that CycB supply to the cytoplasm was more effective than that to the nucleus for rescue from cell-cycle arrest. Ectopic overexpression of NLS-CycB in normal spermatocytes did not affect the meiotic initiation. In other words, these data suggest that reduced cytoplasmic CycB, rather than the accumulation of ectopic nuclear CycB, was involved in the inhibition of meiotic initiation.

### 3.3. Cdk1 Showed Reduced Activation during the Growth Phase and Full-Scale Activation Immediately after Rapid Re-Entry of CycB in the Nucleus upon Meiosis Onset

To further analyze the relationship between Cdk1 activity and CycB localization before and during the onset of meiosis, we performed simultaneous immunostaining with the anti-MPM2 antibody, which can recognize the epitopes phosphorylated by several kinases, including CDK1 [36], and anti-CycB antibody. In normal spermatocytes at the S5 stage (*bam>+*), a faint MPM2 fluorescent signal colocalized with Sa-GFP foci in the nucleolus (Figure 3a,a’). A subtle CycB signal was observed in the cytoplasm at S5 (Figure 3a,a’’). With the shrinkage of the nucleolus visualized by Sa-GFP (Figure 3b,b”’) and the associated phosphorylation of nucleolar proteins by CDK, no detectable MPM2 signal was observed (Figure 3b,b’). The CycB signal in the cytoplasm became more intense at S6, although the N/C ratio of CycB immunofluorescence intensity did not significantly increase (*p* = 0.93) (Figure 3b”,e). In cells showing condensed chromosomes at ProI (Figure 3c””), the N/C ratio slightly increased. The difference was significant (*p* < 0.01). These results indicate that CycB started entering the nucleus from the cytoplasm (Figure 3c”,e). When the CycB signal was detected in the nucleus, MPM2 epitopes appeared in the nucleus (Figure 3c’). Subsequently, the signal intensity sharply increased in the nucleus and cytoplasm from early ProI to prometaphase I (PrometaI) (Figure 3d,d”,e) (*p* < 0.01). This sharp increase in the MPM2 epitopes suggested that the full-scale activation of Cdk1-CycB began and flourished in this period. These observations indicated that full-scale activation of CycB occurred at the nucleus after its rapid re-entry, and CycB spread immediately to the cytoplasm from early ProI to PrometaI (Figure 3e), when Sa-GFP foci were no longer detected (Figure 3d”’). The robust intensity of the anti-MPM2 signal in *Nup62*-silenced cells was not observed in the previous report [36].

### 3.4. CentrosomeSeparation in the Cytoplasm Prior to Full-Scale Cdk1 Activation Depended on Cdk1, Which Was Not Observed in Nup62-Silenced Spermatocytes

Initial activation triggers the full-scale activation of Cdk1 in mitosis [11]. Although we detected faint signals for MPM2 epitopes at the S5 and S6 stages, whether partial activity of Cdk1 was required for certain cellular events at the pre-meiotic stage was unclear. A pair of centrosomes separate from each other before ProI, and Cdk1 is required for this cellular event. In control spermatocytes (*bam>+*), a pair of centrosomes was visualized by immunostaining for Asterless (Asl); centrosomes were localized next to each other at the S5 stage (Figure 4a’). They separated from each other at S6 (Figure 4b’) and reached the opposite poles of cells at ProI (Figure 4c’). The distance between the two centrosomes significantly increased as the cells proceeded from S5 to S6 by twofold (*p* < 0.01) and then to ProI by threefold (*p* < 0.01) (Figure 4h). In contrast, centrosome separation was inhibited in spermatocytes harboring cell-specific *Cdk1RNAi* (Figure 4d,e, yellow bars in 4h), although MPM2 epitopes were not seen on centrosomes (Figure 4a–c). Similarly, separation was also inhibited in *Nup62-*silenced spermatocytes (Figure 4f,g, red bars in 4h). The mean distance between paired centrosomes at S5 in *Cdk1-* or *Nup62-*silenced cells did not significantly differ from that in the controls. In contrast, the distance in *Cdk1*- or *Nup62-*silenced cells at S6 did not significantly increase compared to that in control cells (Figure 4h). Therefore, we concluded that centrosome separation before the onset of meiosis was disrupted in *Nup62-*silenced spermatocytes and in *Cdk1-*silenced cells. These phenotypes suggest that initial low-level activation of Cdk1 was required for centrosome separation and that this activity was inhibited in *Nup62-*silenced spermatocytes.

### 3.5. Cdk1 Was Closely Associated with CAK in the Nucleus during the Growth Phase in Both Normal and Nup62-Silenced Spermatocytes before Meiosis

Next, we explored why Cdk1 was not activated even at a low level in *Nup62-*silenced spermatocytes (*bam>Nup62RNAi*). We investigated the subcellular localization of CAK and assessed its close association with Cdk1. We performed immunostaining on control cells (*bam>+*) using an anti-Cdk7 antibody that specifically recognizes the catalytic subunit of CAK (Figure 5a–c). A fluorescent signal was noticed in the nucleolus of spermatocytes at S5 and subsequently spread over the whole region of the nucleus at S6; no signal was detected at ProI or thereafter (Figure 5a–c). To investigate whether Cdk1 was associated with CAK in spermatocytes, we performed an in situ PLA, which can detect a close association between two proteins if they are close enough to form a complex. The fluorescent signal of Cdk7 was detected in the nuclei of normal cells at the S5 and S6 stages (Figure 5a,b). Consistently, PLA signals were observed only in the nucleus at both stages (Figure 5g,h), suggesting that Cdk1 underwent phosphorylation by CAK in the nucleus. Similarly, both Cdk7 (Figure 5d,e) and PLA signals (Figure 5i,j) were specifically observed in the nuclei of *Nup62-*silenced cells at both stages, as in normal cells. These data suggested that Cdk1 was normally phosphorylated by CAK in the nuclei of *Nup62-*silenced spermatocytes.

### 3.6. The Association of Cdk1 with its Activator Phosphatase, Twine, in the Nucleus and Cytoplasm before Meiosis Did Not Change in Nup62-Silenced Cells

Next, we assessed whether dephosphorylation of Cdk1 at Thr^14^ and Tyr^15^ by Twine phosphatase occurred in *Nup62-*silenced spermatocytes. Since specific antibodies against phosphorylated Thr^14^ and Tyr^15^ of *Drosophila* Cdk1 were not available, we performed immunostaining using anti-Twine antibody to detect its cellular localization before meiosis and subsequently examined its association with Cdk1. We performed immunostaining of control cells (*bam>+*) from S3 to ProI with anti-Twine antibody (Figure 6a–e). A weak fluorescent signal was noticed in the nuclei and cytoplasm of spermatocytes at S3 and S4 (Figure 6a,b), and distinctive signals were predominantly observed in the nucleus from S5 to ProI (Figure 6c–e,c’–e’). Similarly, a fluorescent signal was detected in the nuclei of *Nup62*-silenced spermatocytes at S5 and S6 (Figure 6f–i). The N/C ratio of fluorescence intensity was >1.0 at S5, suggesting that more Twine was localized in the nucleus than in the cytoplasm. We next performed an in situ PLA to detect complex formation between Cdk1 and Twine before meiosis. PLA signals were observed in whole regions of normal spermatocytes at S5 and S6 (Figure 6k–m), indicating that Cdk1 was associated with Twine closely enough to undergo dephosphorylation by the phosphatase in the nucleus and cytoplasm. Similarly, PLA signals were observed in *Nup62*-silenced cells at S5 and S6 (Figure 6n,o), while the signals were not seen using anti-GFP antibody alone (Appendix A). The PLA signals can be seen in both the nucleus and the cytoplasm of the spermatocytes in control and *Nup62RNAi* cells to the extent that the N/C ratio slightly exceeded 1.0 (Figure 6p). These data suggested that Cdk1 activation by Twine took place in whole regions of normal and *Nup62*-silenced cells.

We observed fluorescence of Wee1–GFP (Appendix A) and Myt1–GFP (Appendix A) predominantly in the nucleus and cytoplasm at S5, S6, and ProI. The PLA detected complexes containing Wee1–GFP and Cdk1 (Appendix A) and Myt1–GFP and Cdk1 (Appendix A) in the nucleus and cytoplasm before meiosis and ProI. Based on these results, we concluded that Wee1/Myt1 kinases were associated with Cdk1 during S5–ProI. Next, we investigated whether Polo, an indispensable protein kinase that controls the G2/M transition of the mitotic cycle, is required for meiotic initiation. We noticed many spermatid cysts consisting of 16 cells harboring multiple nuclei in testes harboring *polo*-silenced spermatocytes, suggesting that chromosome segregation took place but that cytokinesis did not occur in either of the meiotic divisions (Appendix A). Consistently, even in *polo*-silenced spermatocytes, MPM2 epitopes appeared from S6 to ProI, as observed in control spermatocytes (Appendix A). Immunofluorescence staining indicated that Polo migrated into the nucleus (Appendix A) slightly earlier than did CycB (Appendix A). Similarly, Polo was localized in the cytoplasm at and before S6 in *Nup62-*silenced cells (Appendix A), whereas CycB was not exported from the nucleus (Appendix A). No detectable differences in the subcellular localization of Polo were noticed in *Nup62-*silenced cells. Moreover, PLA signals indicated complex formation between CycB and Polo–GFP in the nuclei and cytoplasm of control (*bam>+*) (Appendix A) and *Nup62*-silenced cells (Appendix A).

### 3.7. Silencing of the CKI Roughex (Rux), but Not Z600, Rescued the Nup62-Silencing-Induced Inhibition of Meiotic Initiation

Cdk1 was not activated in spermatocytes before the initiation of meiosis when CycB was accumulated in the nucleus owing to the inhibition of nuclear export. No remarkable differences in the association of CAK or Twine with Cdk1 were observed between normal and *Nup62-*silenced cells. Unlike the overexpression of CycB, ectopic expression of constitutively active Cdk1 (Cdk1^T14AY15F^) cannot rescue the inhibition of meiotic initiation in *Nup62-*silenced cells [36]; therefore, we hypothesized that a negative regulator represses Cdk1 activity in the nucleus independently of Cdk1 modification. We investigated whether another type of inhibitor different from Wee1/Myt1 was involved in regulating Cdk1 before meiosis. Two CKIs, Z600/Frs and Rux, bind to CycB–Cdk1 [49,50]. Immunostaining of the spermatocytes in normal testes (*bam>+*) using an anti-Z600 antibody and observation of spermatocytes expressing HA-tagged Z600 revealed that Z600 was localized in the nuclei of spermatocytes at S5 and ProI (Appendix A). At the S6 stage, a distinct fluorescent signal was predominantly observed in the nucleus, with a less intense signal in the cytoplasm (Appendix A). These signals were almost lost in *Z600-*silenced spermatocytes (Appendix A). However, Z600 depletion did not influence the meiotic phenotype of *Nup62-*silenced cells. Rux is a negative regulator of the CycA–Cdk1 complex; however, it can bind to CycB–Cdk1 to suppress its kinase activity [49]. We investigated whether Rux was involved in repressing Cdk1 before meiosis. In testes harboring *Nup62-*silenced spermatocytes, all spermatid cysts (*n* = 39 cysts) consisted of 16 cells (Figure 7b). In contrast, testes harboring *rux-*silenced spermatocytes contained abnormal spermatid cysts. Although they consisted of 64 spermatids, 59 out of 70 cells harbored small and multiple nuclei (Figure 7c). This meiotic phenotype was consistent with the reported phenotype of hypomorphic *rux* mutants that originated owing to an extra round of chromosome segregation without cytokinesis [51]. In testes containing spermatocytes featuring the simultaneous silencing of *Nup62* and *rux* (*bam>Nup62RNAi*, *ruxRNAi*), we frequently observed spermatid cysts consisting of 16 cells, and some of the spermatids contained multiple small nuclei (arrow in Figure 7d), indicating that chromosome segregation occurred without cytokinesis (100/109 cells). Other spermatids contained nuclei larger than those of normal spermatids, indicating that meiosis did not occur, as seen in the spermatid cysts of *Nup62*-silenced testes (arrowhead in Figure 7d) (9/109 cells). These phenotypes suggested that *rux* silencing rescued the inhibition of meiotic initiation in *Nup62-*silenced spermatocytes. Consistent with this finding, we observed several spermatocytes with intense MPM2 signals in the nuclei and cytoplasm of matured spermatocytes harboring *ruxRNAi* and *Nup62RNAi* (Figure 7i’,j) (10% (31/303 cells)). The frequency of MPM2-positive cells among the cells harboring the simultaneous silencing was equivalent to that in the control cells. The number of MPM2-positive spermatocytes increased in testes harboring *rux-*silenced spermatocytes (46/87 cells). These results suggest that *rux* silencing can rescue meiotic cell-cycle arrest in *Nup62-*silenced spermatocytes.

### 3.8. Depletion of Importin β Inhibited Rapid Nuclear Re-Entry of CycB and Initiation of Meiosis but Did Not Affect Centrosome Separation before Meiosis

Upon nuclear import of CycB in the pre-meiotic spermatocytes treated with LMB, the protein was relatively slowly imported into the nucleus. It took over an hour for CycB signals in the nucleus and cytoplasm to become almost equal (Figure 1i). In contrast, it took only 6–7 min at the onset of meiosis (Figure 8a,b). To identify the factors required for the rapid nuclear import of CycB at the onset of meiosis, we investigated whether any nucleocytoplasmic transport receptor that imports and exports proteins through the nuclear pores was involved in the nuclear import of CycB. We silenced *Fs(2)Ket*, encoding *Drosophila* importin β, using RNAi. Spermatocyte-specific silencing of importin β revealed that all spermatid cysts at the onion stage consisted of 16 cells (12/12 cysts), in which three clusters of condensed chromosomes were still observed (Appendix A). The chromatin organization in the spermatids was different from that in *Nup62-* or *emb*-silenced spermatids, in which under-condensed chromatin was present in the nucleus (Appendix A). Therefore, meiotic initiation was inhibited in *Fs(2)Ket-*silenced spermatocytes, and the cell cycle was arrested at different stages immediately before the full-scale activation of Cdk1. To clarify this, we investigated the cellular localization of CycB. In control testes (*bam>+*), we observed robust CycB–GFP fluorescence in the cytoplasm of spermatocytes at the S5 stage; however, no fluorescence was noticed in the nuclei of spermatids at the onion stage (Figure 8c’,d’). CycB–GFP fluorescence was observed in the nuclei of *Nup62-*silenced spermatocytes (*bam>Nup62RNAi*) at the same stage (Figure 8e’), and a fluorescent signal was sustained in the nuclei of spermatids harboring Nebenkerns (arrow in Figure 8f). In contrast, CycB–GFP was localized in the cytoplasm of *Fs(2)Ket*-depleted cells (Figure 8g’,h’). These cytological phenotypes and cellular localization of CycB are consistent with the interpretation that *Fs(2)Ket* is required for the nuclear import of CycB–Cdk1 to complete full-scale Cdk1 activation. *Fs(2)Ket* depletion did not influence centrosome separation, which occurred before the full-scale activation of Cdk1 in meiosis (Figure 8k–m). Unlike that in *Cdk1-* or *Nup62-*silenced cells (Figure 4d’–g’), centrosome separation in *Fs(2)Ket-*silenced spermatocytes was indistinguishable from that of the control cells (Figure 8i’–l’). Therefore, centrosome separation was not affected by the inhibition of CycB re-entry to the nucleus.

## 4. Discussion

The Cdk1-CycB complex serves as a common master regulator of the cell-cycle progression into the M phase in mitosis and meiosis. The importance of spatial and feedback regulation in the activation of Cdk1 has been well demonstrated in mitosis. In contrast, the regulatory mechanism in meiotic initiation remained unclear. A previous study reported that Cdk1 is not activated, and that meiosis does not initiate when the export of the kinase complex is inhibited from the nucleus. We aimed to clarify the importance of the subcellular localization of Cdk1-CycB in determining when meiosis is initiated in *Drosophila* males. We performed a time-lapse observation of living pre-meiotic spermatocytes and found that Cdk1–CycB continuously shuttled into and out of the nucleus before meiosis. Overexpression of CycB, but not that of NLS-CycB, rescued reduced cytoplasmic CycB and inhibition of meiosis in *Nup62*-silenced cells. Furthermore, we investigated the subcellular localization of positive regulators and showed that the Cdk1 interacted with Cdk-activating kinase and Twine/Cdc25C even in the nuclei of *Nup62-*silenced cells, suggesting that other regulatory factors were involved in the failure of meiotic initiation. Silencing of one of the CKIs, *roughex*, rescued the Cdk1 inhibition and initiated meiosis. Full-scale Cdk1 activation occurred in the nucleus shortly after its rapid nuclear entry. The complex re-entered the nucleus via importin β at the onset of meiosis.

### 4.1. Loss of Cytoplasmic CycB and Concomitantly Reduced Initial Cdk1 Activity Inhibit Meiosis in Nup62-Silenced Spermatocytes When Nuclear Export Is Disrupted

Cdk1 activation is an essential step for initiating mitotic and meiotic divisions. Depletion of the Nup62 subcomplex of the NPC inhibits Cdk1 activation and meiotic initiation. CycB is accumulated in the nuclei of *Nup62-*silenced cells, whereas it is localized in the cytoplasm of normal cells before meiotic initiation [36]. We explored why CycB–Cdk1 was localized in the nucleus upon downregulating the expression of the Nup62 subcomplex or exportin orthologue. After the growth phase in normal spermatocytes, CycB–Cdk1 migrated to the nucleus, and meiosis started. Taking these results together with the previous findings that the full-scale activation of Cdk1 was suppressed in *Nup62*-silenced cells [36], we speculated that abnormal subcellular localization may be involved in inhibiting Cdk1 activation in *Nup62-*silenced cells. CycB should be transported to the nuclei of spermatocytes in advance during early stages of the growth phase. CYCB1 continuously shuttles in and out of the nucleus before the M phase in human cells [52]. Similarly, we observed that CycB was transported to the nucleus and then immediately exported from the nucleus during the growth phase in normal spermatocytes. Two reasons for cell-cycle arrest following *Nup62* silencing before the initiation of meiosis may be considered: (1) the precocious localization of CycB–Cdk1 at the nucleus as a consequence of *Nup62* silencing may dominantly inhibit the activation of endogenous Cdk1 before the onset of meiosis; or (2) the reduced amount of Cdk1–CycB in the cytoplasm of *Nup62-*silenced cells may inhibit meiotic initiation. Ectopic overexpression of CycB, but not NLS-CycB, rescued the inhibition of meiotic initiation in *Nup62-*silenced cells. These observations support the second possibility, that cytoplasmic CycB is more important than nuclear CycB for Cdk1 activation.

The absence of MPM2 epitopes in the nuclei and cytoplasm of the silenced cells indicated that Cdk1 was not activated in either compartment. Consistently, Cdk1-dependent events, such as centrosome separation, were suppressed in the cytoplasm, suggesting that initial Cdk1 activation takes place in the cytoplasm. Therefore, CycA–Cdk1 can be considered a regulator of initial activation. *Drosophila* CycA functions as a mitotic cyclin, unlike its mammalian orthologue. It is translated earlier than CycB during the growth phase [28]. In spermatocytes expressing constitutively active Cdk1, which is not suppressed by Myt1, and in *myt1* hypomorphic mutant cells, premature centriole disengagement occurs. This meiotic phenotype can be suppressed by the depletion of CycA activity [53]. These previous results suggest that CycA–Cdk1 activity can influence centrosome dynamics in male meiosis. We have observed that CycA depletion results in the accumulation of CycB in the nucleus during the G2 phase and the inhibition of full-scale Cdk1 activation (Tanaka, Y., Yamazoe, K., and Inoue, Y.H., manuscript in preparation). Further studies are necessary to clarify the possible role of CycA–Cdk1 in meiotic initiation.

### 4.2. Cdk1 Activation May Be Suppressed in the Nuclei of Spermatocytes by Roughex until the Onset of Meiosis

Male meiosis does not initiate until Cdk1–CycB is exported from the nucleus to the cytoplasm during the growth phase. The inhibition of meiosis in *Nup62-*silenced spermatocytes may be not responsible for the aberrant dephosphorylation of Tyr^14^ and Thr^15^ residues in Cdk1 by Twine, as previously reported, nor for the aberrant phosphorylation of Thr^161^ by CAK in the previous study [36]. The data suggesting colocalization and close association of CAK with Cdk1, as well as colocalization and close association of Twine with Cdk1, in the *Nup62RNAi* spermatocytes are consistent with a model in which CAK activates Cdk1 in the nuclei of *Nup62*-depleted cells. Even so, they do not directly prove that Cdk1 was phosphorylated and dephosphorylated by the kinase and phosphatase, respectively. Further biochemical experiments are needed to confirm the phosphorylation status of the relevant amino acid residues within Cdk1, although specific antibodies that recognize the phosphorylated peptides in *Drosophila* are not available. If the speculation above is correct, mechanisms other than protein modification may suppress Cdk1 activation. CKIs directly bind to cyclin–Cdk complexes to suppress their activities and regulate cell-cycle progression [8]. Our genetic analysis suggested that Rux may be involved in inhibiting Cdk1 activation until CycB–Cdk1 is exported from the nucleus to the cytoplasm. Two possible mechanisms by which Rux could suppress Cdk1–cyclins in the nucleus can be considered. First, Rux may inhibit Cdk1–CycA activity in the nucleus. Since CycA is translated and activated earlier than CycB [27,47,53], Cdk1–CycB activity, a major driver of male meiosis, may be suppressed via the inhibition of Cdk1–CycA by Rux until the complex is exported from the nucleus. Alternatively, Rux may directly suppress Cdk1–CycB in spermatocytes before meiosis. Rux can bind to CycB–Cdk1 and suppress its CycB-dependent kinase activity [49]. However, the effects of Rux on mitotic cyclin–Cdk1 complexes open up the possibility that it also contributes to the regulation of mitotic initiation in *Drosophila* embryos [54]. Whether *rux* is involved in determining the timing of male meiosis should be investigated.

### 4.3. Rapid Re-Entry of Cdk1–CycB to the Nucleus May Play an Important Role in the Full-Scale Activation of Cdk1 with Initial Activity and Meiotic Initiation

Our observations indicate that the nuclear re-entry of CycB is a rapid process. The nuclear transfer machinery may be activated by Cdk1, thereby enabling rapid nucleus-to-cytoplasm transport. Mammalian CYCB1 is imported through direct interaction with importin β. Cdk1 phosphorylates importin β, stimulating an interaction between importins α and β to accelerate protein transport [55,56,57]. We noticed that importin β was involved in the rapid nuclear import of CycB, although a typical NLS was not identified in *Drosophila* CycB. Importin β was not required for slow import in the G2 phase before centrosome separation, as the event was not affected in *Fs(2)-*silenced cells. Polo-like kinase suppresses the nuclear export of cyclin B1–Cdk1 via phosphorylation of the nuclear export signal of cyclin in animal cells [11,58]. This kinase is thought to facilitate the rapid accumulation of CycB in the nucleus. However, *polo* may not play a critical role in the rapid nuclear import of Cdk1–CycB at the onset of *Drosophila* male meiosis because the silencing of *polo* did not affect meiotic initiation and the nuclear export of Cdk1–CycB did not change in *Nup62-*silenced cells.

We obtained evidence that the subcellular localization of essential cell-cycle regulators plays an important role in Cdk1 activation and meiotic initiation. Cdk1 needs to be activated in the cytoplasm during the G2/M transition; otherwise, meiosis cannot initiate properly. When Cdk1 remains in the nucleus, the level of Cdk1–CycB is reduced in the cytoplasm, becoming insufficient to initiate meiosis. If the positive regulators required for Cdk1 activation are localized to the cytoplasm, Cdk1 must be exported to the cytoplasm for activation. Conversely, if negative regulators are localized to the nucleus, they need to be released from Cdk1 for its activation. CAK and Twine were localized in the nucleus throughout the growth phase. Subcellular localization of these positive factors does not support the first possibility. In contrast, the negative regulators Wee1/Myt1 were also predominantly localized in the nucleus. Rux is localized in the cytoplasm when CycA re-enters the nucleus [53]. Before this developmental stage, the subcellular localization of Rux was not reported. However, we observed that *rux* silencing rescued the accumulation of CycB–Cdk1 in the nucleus, thereby suggesting that the Cdk1 complex was suppressed by Rux until it was released from the inhibitor.

Cell-cycle regulation differs in some ways between *Drosophila* male meiosis and mitosis. Before the initiation of mitosis in animal cells, CycB migrates to the nucleus to avoid premature mitosis until DNA damage checkpoints are verified [59]. In contrast, the initiation of meiosis may not be permitted until the clearance of further conditions that the pre-meiotic spermatocytes should fulfill, for example, by ensuring sufficient cell growth. Several proteins and mRNAs are required for meiotic division, and post-meiotic events occur during the growth phase [23,60]. In a hypomorphic mutant for *eIF4G* encoding a eukaryotic translation initiation factor, the growth of germline cells was inhibited. Moreover, neither meiosis nor sperm differentiation was observed in mutant testes [61,62]. Therefore, Cdk1 activation that terminates the growth phase may need to be strictly regulated before meiosis, for example, through additional checkpoints that monitor cell growth.

### 4.4. Stepwise Activation of Cdk1 Is Associated with Nuclear–Cytoplasmic Shuttling of CycB Mediated by the Nup62 Subcomplex, Exportin, and Importin β

We propose a new model regarding the stepwise activation of Cdk1–cyclins associated with the nuclear–cytoplasmic shuttling of CycB (Figure 9). During a prolonged G2 phase in spermatocytes, Cdk1–CycB continues to be modified by Wee1/Myt1 and Twine in the cytoplasm and nucleus. The complex has an intrinsic ability to temporally migrate to the nucleus. Simultaneously, it is exported more rapidly back to the cytoplasm through a unique exportin orthologue, Emb, via the Nup62 subcomplex of the NPC. Most kinase complexes are inactivated by Wee1/Myt1, which initially dominates over Twine, and are further suppressed by Rux through the suppression of Cdk1–cyclins (CycA or CycB) in the nucleus. Nevertheless, a small population of Cdk1–cyclins may execute some pre-meiotic events such as centrosome separation. With a sharp increase in CycB expression shortly before the onset of meiosis, a small portion of active Cdk1 initiates the production of a large amount of the active Cdk1 complex through the activation of Twine and inactivation of negative regulators. The kinase complex is rapidly imported into the nucleus via the Fs(2)Ket/importin β-mediated pathway. Through positive and double-negative feedback loops, the resultant CycB–Cdk1 triggers meiotic initiation after the completion of full-scale activation in the nucleus. Further investigations are warranted to validate this model. In this study, we investigated Cdk1 activation using MPM2 antibody, which recognized phosphorylated proteins by several kinases, including Cdk1 [63]. Although we can conclude that Cdk1 was not activated either when no MPM2 epitopes were observed, the detection of the epitopes does not necessarily prove that Cdk1 was activated. If further experiments using another specific probe that can detect the activation of this kinase directly were to become available [64], a more reliable conclusion could be reached. This is one of the limitations of this study and a challenge for the future. We demonstrated that colocalization and close association of CAK with Cdk1, as well as colocalization and close association of Twine with Cdk1, could be normally observed even when the nuclear export of CycB-Cdk1 was inhibited. However, these data do not directly prove that Cdk1 was phosphorylated and dephosphorylated by the kinase and phosphatase, respectively. However, the specific antibodies that recognize the phosphorylated peptides in *Drosophila* Cdk1 are not available. That is another limitation of our current study. Further biochemical experiments, such as comprehensive identification of phosphorylated residues in Cdk1 by phosphoproteomics, are needed to confirm the phosphorylation status of Cdk1. This study provided evidence suggesting that initiation of *Drosophila* male meiosis requires stepwise activation of Cdk1-CycB by altering its subcellular localization through NPC-mediated nuclear-to-cytoplasmic transport. This finding does not only help to elucidate the initiation mechanism of male meiosis but may also have implications for understanding the mechanism of *Drosophila* female meiosis, meiosis in other organisms, and the regulation of mitosis in various tissues.

## Figures and Tables

**Figure 1 cells-12-02611-f001:**
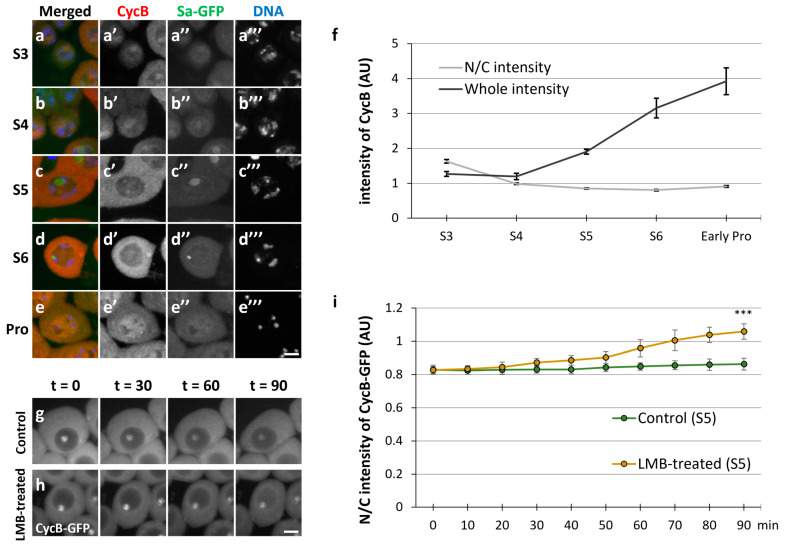
Nuclear–cytoplasmic shuttling of cyclin B in spermatocytes during the growth phase. (**a**–**e**) Anti-Cyclin B (CycB) immunostaining of spermatocytes with anti-cyclin B (CycB) antibody from the growth phase (S3–S6 in (**a**–**d**)) to the onset of meiosis (prophase I (Pro) in (**e**)). Images show CycB immunofluorescence (red in (**a**–**e**), white in (**a**’–**e**’)), Sa-GFP fluorescence for visualizing the nucleolus (green in (**a**–**e**), white in (**a**’’–**e**’’)), and DNA staining with 4′,6-diamidino-2-phenylindole (DAPI) (blue in (**a**–**e**), white in (**a**’’’–**e**’’’)). Scale bar: 10 μm. (**f**) Quantification of CycB immunofluorescence intensity in whole spermatocytes (black line) and nuclei relative to that in the cytoplasm of spermatocytes (gray line) at each stage represented in (**a**–**e**). Intensities of the immunofluorescence in spermatocytes at the S3 to ProI phases were individually measured in whole cell regions, the cytoplasm, and the nucleus. The mean intensity value in the whole region was calculated at each stage, displayed on the *y*-axis (arbitrary unit: AU), and drawn with a black line. The ratio of the intensity in the cytoplasm to that in the nucleus (N/C intensity) was displayed by a gray line. Data represent means ± 95% confidence intervals (CIs) (>16 cells). (**g**,**h**) Time-lapse observation of CycB in the growth phase (S5) of living spermatocytes expressing CycB–GFP and treated with leptomycin B (LMB) (**h**) and that of untreated cells (**g**). Ninety images were captured every minute. Four selected images of a live spermatocyte are presented. Scale bar: 10 μm. (**i**) Comparative quantification of nuclear GFP fluorescence intensity relative to cytoplasmic intensity in LMB-treated (orange line) and untreated (green line) spermatocytes expressing CycB–GFP. The intensities of CycB-GFP in the whole regions of the cells at the time indicated by t = 0 to 90 min were measured. The ratio of the intensity in the cytoplasm to that in the nucleus (N/C intensity) was displayed on the *y*-axis (with a green line (control cells at S5) and an orange line (the cells treated with LMB)). Data represent means ± 95% CIs (11 untreated and 8 LMB-treated spermatocytes). Significance was tested between control cells and LMB-treated cells at the last time point (*t* = 90). *** *p* < 0.001 (Mann–Whitney test).

**Figure 2 cells-12-02611-f002:**
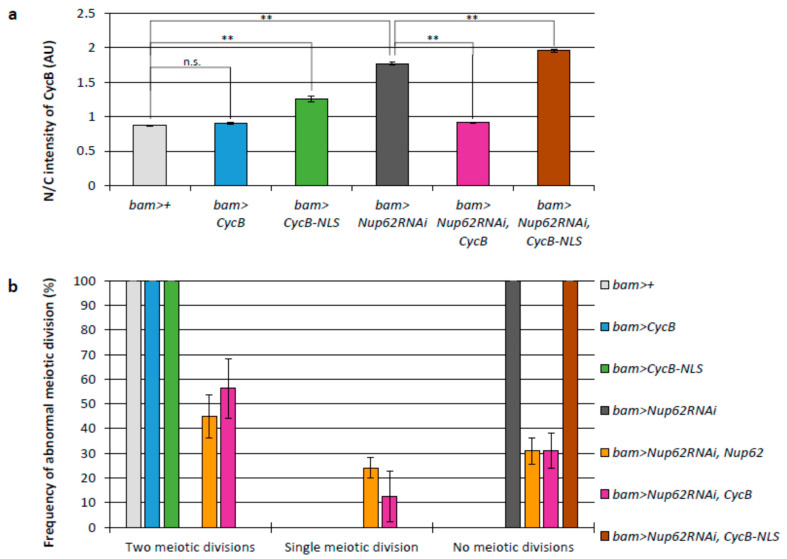
Effects of cytoplasmic CycB on the meiotic defect of *Nup62*-depleted spermatocytes. (**a**) Quantification of the anti-CycB immunofluorescence intensity in the nucleus relative to that in the cytoplasm of growth-phase spermatocytes (S5) of the following genotypes: *bam>+* (light gray column), *bam>CycB* (blue column), *bam>CycB-NLS* (green column), *bam>Nup62RNAi* (dark gray column), *bam>Nup62RNAi*, *CycB* (magenta column), *bam>Nup62RNAi*, *CycB-NLS* (brown column). The ratio of the fluorescence intensity in the cytoplasm to that in the nucleus (N/C intensity) was displayed on the *y*-axis. Data are presented as means ± SEMs (*n* > 15). ** *p* < 0.01; n.s.: not significant; one-way ANOVA followed by Bonferroni’s post hoc comparison. (**b**) Frequency of spermatid cysts formed through two meiotic divisions (normal), one meiotic division (abnormal), or no meiotic divisions (abnormal). More than 15 spermatid cysts were collected from each of the 24 testes of the genotypes referred to in (**a**), in addition to another control, *bam>Nup62RNAi*, *Nup62* (orange column). Data are presented as means ± SEMs.

**Figure 3 cells-12-02611-f003:**
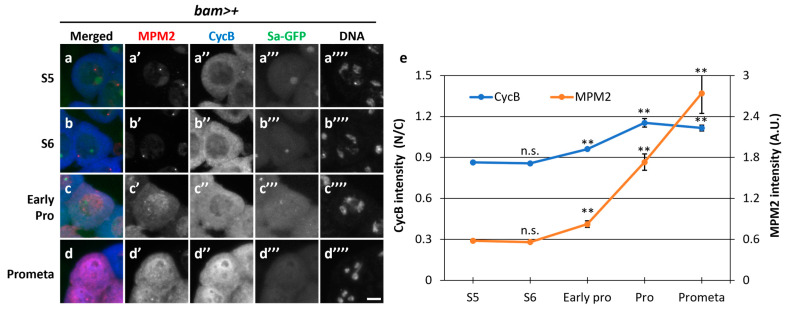
Timing of full-scale Cdk1–CycB activation and nuclear re-entry of CycB at the onset of meiotic division. (**a**–**d**) Fluorescent images of mature spermatocytes co-immunostained with anti-MPM2 and -CycB antibodies before (**a**,**b**) and immediately after the onset of meiosis (**c**) and during meiosis (**d**). Images show anti-MPM2 (red in (**a**–**d**), white in (**a**’–**d**’)) and anti-CycB (blue in (**a**–**d**), white in (**a**’’–**d**’’) immunofluorescence, Sa-GFP fluorescence for visualizing the nucleolus (green in (**a**–**d**), white in (**a**’’’–**d**’’’)), and DNA staining with DAPI (white in (**a**’’’’–**d**’’’’)). Scale bar: 10 μm. (**e**) Quantification of anti-MPM2 and nuclear anti-CycB immunostaining intensities at each stage represented in (**a**–**d**). The anti-CycB and anti-MPM2 immunofluorescence intensities of the spermatocytes at S5 to prometaphase I (Prometa) were measured. The ratio of the anti-CycB intensity in the cytoplasm to that in the nucleus (N/C intensity) was displayed by a blue line. The fluorescence of the MPM2 epitopes (AU) is displayed by an orange line. Analyses were performed on 71 cells in S5, 133 cells in S6, 30 cells in early prophase, 24 cells in prophase, and 7 cells in prometaphase. Data are presented as means ± SEMs. Significance was tested using one-way ANOVA followed by Bonferroni’s post hoc comparison tests. ** *p* < 0.01, n.s.: not significant.

**Figure 4 cells-12-02611-f004:**
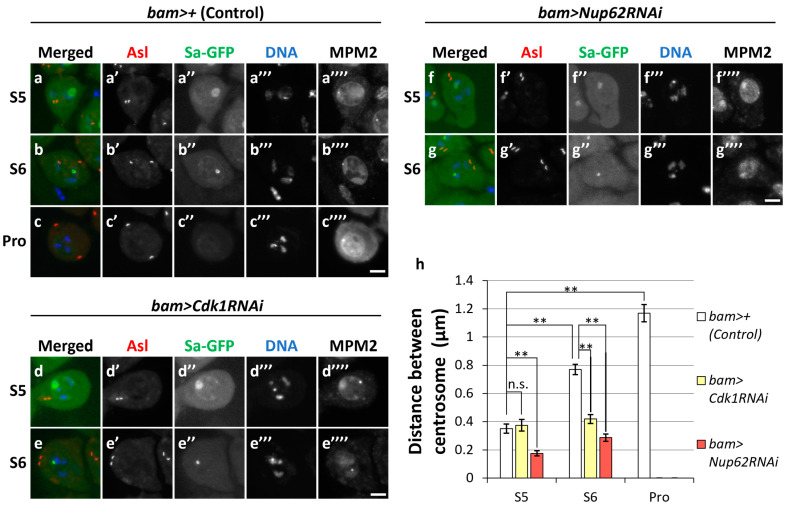
Cdk1-mediated centrosome separation before the initiation of meiotic division. (**a**–**g**) Observation of mature spermatocytes (growth phase at S5 and S6 in (**a**,**b**,**d**–**g**); at prophase I (Pro) in (**c**)) immunostained with anti-Asl and anti-MPM2 antibodies. Normal control (*bam>+*) (**a**–**c**), *Cdk1*-silenced (**d**,**e**), and *Nup62*-silenced (**f**,**g**) cells. Images show anti-Asl immunofluorescence to visualize centrosomes (red in (**a**–**g**), white in (**a**’–**g**’)), Sa-GFP fluorescence for visualizing the nucleolus (green in (**a**–**g**), white in (**a**’’–**g**’’)), and DNA staining with DAPI (blue in a-g, white in (**a**’’’–**g**’’’)) and anti-MPM2 epitopes (white in (**a**’’’’–**g**’’’’)). Scale bar: 10 μm. (**h**) Quantification of the distance between paired centrosomes in S5 to ProI spermatocytes. The lengths between paired centrosomes were measured in each spermatocyte and the mean length was displayed as a white bar (control cells, *bam>+*), a yellow bar (*Cdk1RNAi*; *bam>Cdk1RNAi*), or a red bar (*Nup62RNAi*; *bam>Nup62RNAi*). Data are presented as means ± SEMs (*n* > 29 cells). No ProI cells were observed in *bam>Cdk1RNAi* or *bam>Nup62RNAi* testes. Significance was tested by one-way ANOVA followed by Bonferroni’s post hoc comparison tests. ** *p* < 0.01. n.s.; not significant.

**Figure 5 cells-12-02611-f005:**
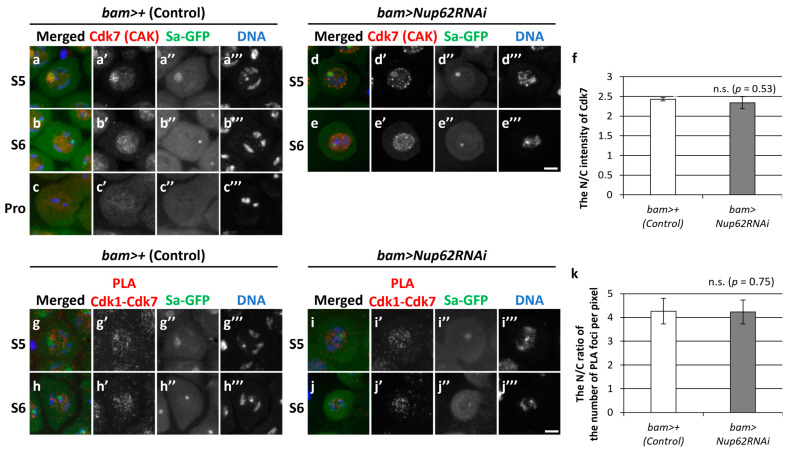
Intracellular localization of Cdk7 and in situ PLA results to monitor a close association between Cdk1 and Cdk7 in spermatocytes at S5, S6, and Prophase I (Pro). (**a**–**e**) Anti-Cdk7 immunostaining of normal (**a**–**c**) and *Nup62*-silenced (**d**,**e**) spermatocytes in the growth phase ((**a**,**d**) at S5; (**b**,**e**) at S6) and prophase (Pro) (**c**). Images show anti-Cdk7 immunofluorescence (red in (**a**–**e**), white in (**a**’–**e**’)), Sa-GFP fluorescence for visualizing the nucleolus (green in (**a**–**e**), white in (**a**’’–**e**’’)), and DNA staining with DAPI (blue in (**a**–**e**), white in (**a**’’’–**e**’’’)). Scale bar: 10 μm. (**f**) Ratio of the intensity of anti-Cdk7 immunofluorescence in the nucleus to that in the cytoplasm. The fluorescence intensity of the spermatocytes at S5 to Pro was measured. The mean ratio of the intensity in the cytoplasm to that in the nucleus (N/C intensity) was displayed as a white bar (control cells) or a gray bar (*Nup62RNAi* cells). Data are presented as means ± 95% CIs (*n* >16 cells for each genotype). Significance was tested using the Mann–Whitney test; n.s.: not significant. (**g**–**j**) In situ PLA to detect the close interaction between Cdk1 and Cdk7 in normal (**g**,**h**) and *Nup62*-silenced (**i**,**j**) spermatocytes at S5 (**g**,**i**) and S6 (**h**,**j**). Scale bar: 10 μm. (**k**) Ratio of the number of PLA signals in the nucleus to that in the cytoplasm. The number of PLA-positive foci in the nucleus or cytoplasm of each spermatocyte at the S5 stage was counted. The number per pixel in each compartment was calculated. The ratio of the number in the cytoplasm to that in the nucleus (N/C intensity) was displayed on the *y*-axis. Data are presented as means ± 95% CIs (*n* > 300 cells). Significance was tested using the Mann–Whitney test.

**Figure 6 cells-12-02611-f006:**
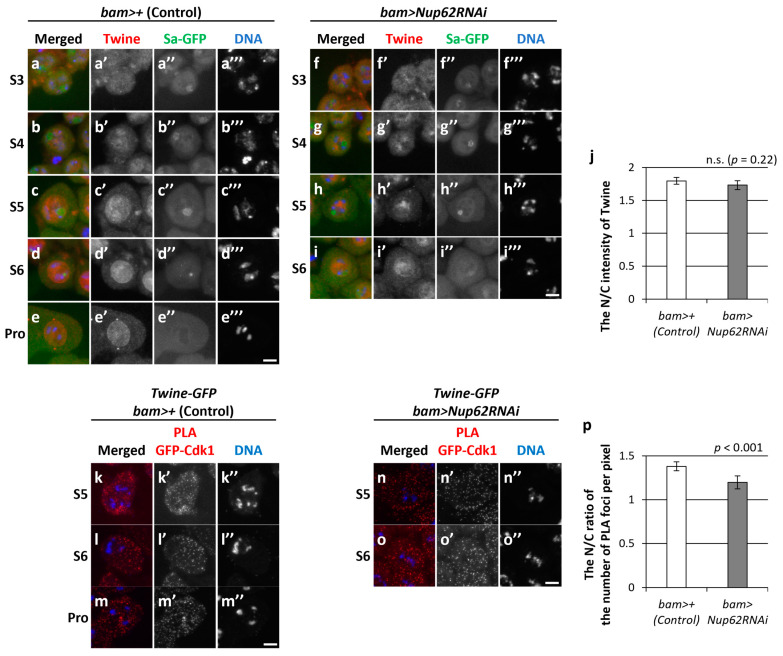
Intracellular localization of Twine and in situ PLA to monitor a close association between Cdk1 and Twine. (**a**–**i**) Anti-Twine immunostaining of normal (**a**–**e**) and *Nup62*-depleted (**f**–**i**) spermatocytes from S3 to S6 of the growth phase (**a**–**d**,**f**–**i**) and prophase I (Pro) (**e**). Images show anti-Twine immunofluorescence (red in (**a**–**i**), white in (**a**’–**i**’)), Sa-GFP fluorescence for visualizing the nucleolus (green in (**a**–**i**), white in (**a**’’–**i**’’)), and DNA staining with DAPI (blue in (**a**–**i)**, white (**a**’’’–**i**’’’)). Scale bar: 10 μm. (**j**) Ratio of the intensity of anti-Twine immunofluorescence in the nucleus to that in the cytoplasm. The fluorescence intensity of the spermatocytes at S5 was measured. The mean ratio of the intensity in the cytoplasm to that in the nucleus (N/C intensity) was displayed as a white bar (control cells) or a gray bar (*Nup62RNAi* cells). Data are presented as means ± 95% Cis (*n* > 18 cells). Significance was tested by the Mann–Whitney test. (**k**–**o**) In situ PLA to detect the close interaction between Cdk1 and Twine in normal (**k**–**m**) and *Nup62*-silenced (**n**,**o**) spermatocytes at the pre-meiotic stage (**k**,**n** at S5; **l**,**o** at S6) and prophase I (Pro) (**m**). Scale bar: 10 μm. (**p**) Ratio of the number of PLA signals in the nucleus to that in the cytoplasm. The number of PLA-positive foci in the nucleus or cytoplasm of each spermatocyte at the S5 stage was counted. The number per pixel in each compartment was calculated. The mean ratio of the number in the cytoplasm to that in the nucleus (N/C intensity) was displayed as a white bar (control cells) or a gray bar (*Nup62RNAi* cells). Data are presented as means ± 95% CIs (*n* > 300 cells). Significance was tested by the Mann–Whitney test. n.s.: not significant.

**Figure 7 cells-12-02611-f007:**
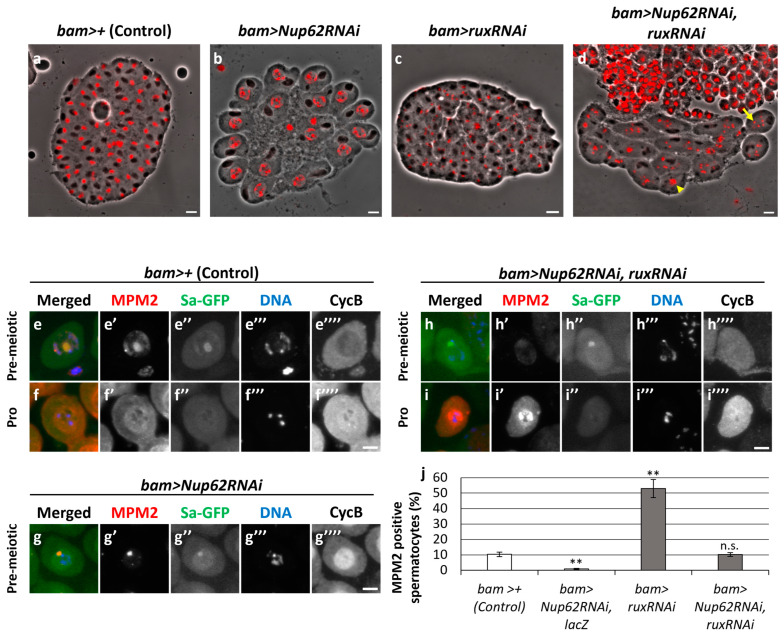
Rescue of the phenotypes of *Nup62RNAi* spermatocytes, including the failure of meiotic division, loss of anti-MPM2 immunostaining in the cytoplasm and nucleus, and nuclear accumulation of CycB by simultaneous depletion of *rux* encoding a CKI. (**a**–**d**) Phase-contrast micrographs of a single intact cyst composed of spermatids at the onion stage after the completion of meiosis II or at a slightly later stage. (**a**) Normal control spermatid cyst containing 64 cells. (**b**) An intact spermatid cyst composed of 16 spermatids derived from *Nup62RNAi* spermatocytes without meiotic divisions. The spermatids harbor under-condensed chromatin in the nuclei. (**c**) An intact spermatid cyst containing > 64 nuclei with a smaller size and >64 Nebenkerns generated from *ruxRNAi* spermatocytes. (**d**) An intact-like spermatid cyst containing >16 Nebenkerns and multiple smaller nuclei (arrow) together with larger-sized nuclei (arrowhead), which were derived from spermatocytes harboring *Nup62RNAi* and *ruxRNAi*, suggesting that meiotic divisions occurred in some spermatocytes of the cyst. Nuclei stained with DAPI are colored in red. Round or slightly extended black structures adjacent to nuclei correspond to Nebenkerns, mitochondrial derivatives. Scale bar: 10 μm. (**e**–**i**) Anti-MPM2 immunostaining (red in (**e**–**i**), white in (**e**’–**i**’)) of spermatocytes at the pre-meiotic S5 or an earlier stage to monitor Cdk1 activation. Sa-GFP (green in (**e**–**i**), white in (**e**”–**i**”)) is a marker to identify the stage of the growth phase. DNA staining (blue in (**e**–**i**), white in (**e**”’–**i**”’)); anti-CycB immunostaining (white in (**e**””–**i**””)). Scale bar: 10 μm. (**e**,**f**) Control spermatocytes at the pre-meiotic stage (**e**) and ProI (**f**). (**g**) Spermatocytes harboring testis-specific depletion of *Nup62* (*bam>Nup62RNAi*) at the pre-meiotic stage. (**h**,**i**) Spermatocytes harboring testis-specific simultaneous depletion of *Nup62* and *rux* (*bam>Nup62RNAi*, *ruxRNAi*) at the pre-meiotic stage (**h**) and ProI (**i**). Stages of spermatocytes were determined using the Sa-GFP marker. (**j**) Frequencies of spermatocytes harboring MPM2 epitopes in control (*bam>+*), *Nup62RNAi* (*bam>Nup62RNAi*, *LacZ*), *ruxRNAi* (*bam>ruxRNAi*), and *Nup62RNAi* and *ruxRNAi* (*bam>Nup62RNAi*, *ruxRNAi*) spermatocytes. ** *p* < 0.01, n.s.: not significant; one-way ANOVA followed by Bonferroni’s post hoc comparison. Error bars indicate SEMs.

**Figure 8 cells-12-02611-f008:**
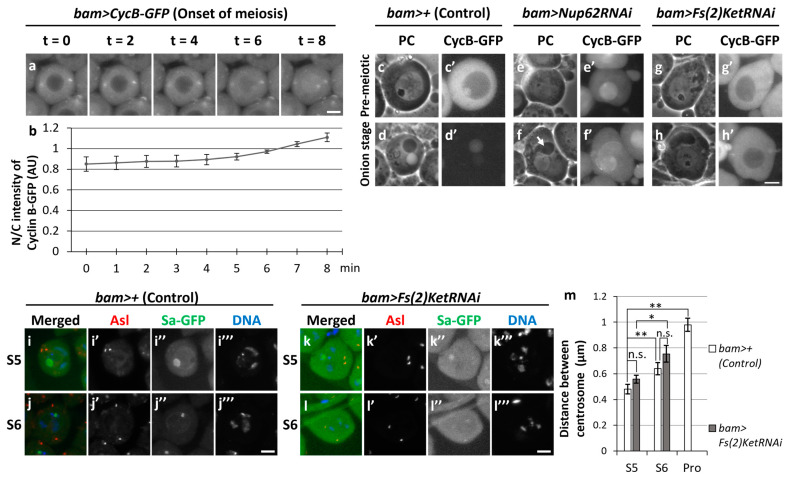
Silencing of *Fs(2)Ket* resulted in the failure of meiotic divisions in spermatocytes and subsequent spermatid differentiation without meiosis. (**a**) Time-lapse observation of CycB–GFP fluorescence in living mature spermatocytes at the onset of meiosis. Ten images were captured every minute from the timepoint when a pair of centrosomes reached opposite poles (*t* = 0). Four selected images of a living spermatocyte are presented. Scale bar: 10 μm. (**b**) Quantification of nuclear fluorescence intensity of GFP relative to that in the cytoplasm of spermatocytes. The fluorescence intensities in the spermatocytes at the time indicated by t = 0 to 8 min were measured in whole cell regions. The mean ratio of the intensity in the cytoplasm to that in the nucleus (N/C intensity) was displayed by a solid line. Data represent means ± 95% CIs (10 spermatocytes). (**c**–**h**) Fluorescence micrographs of living spermatocytes or spermatids. (**c**’–**h**’) CycB–GFP fluorescence in pre-meiotic spermatocytes harboring nucleoli (**c**,**e**,**g**) and onion-stage spermatids (**d**,**f**,**h**). Relatively dark spheres in phase-contrast micrographs of spermatocytes correspond to nucleoli. A white or light sphere and darker round or ellipsoid forms represent the nucleus and Nebenkerns, respectively. (**c**,**d**) Normal control cells (*bam>+*). (**e**,**f**) *Nup62RNAi* cells (*bam>Nup62RNAi*). (**g**,**h**) *Fs(2)KetRNAi* cells (*bam>Fs(2)RNAi*). Growth stages of cells were determined according to the size and morphology of the nucleolus. Scale bar: 10 μm. (**i**–**l**) Anti-Asl immunostaining (red in (**i**–**l**), white in (**i**’–**l**’)) of spermatocytes at S5 (**i**,**k**) and S6 (**j**,**l**). (**i**,**j**) Control spermatocytes (*bam>+*). (**k**,**l**) Spermatocytes harboring *Fs(2)KetRNAi* (*bam>Fs(2)RNAi*). Growth stages were determined using Sa-GFP fluorescence. (**m**) Quantification of the distance between paired centrosomes of spermatocytes from S5 to ProI. The length between paired centrosomes were measured in each spermatocyte and the mean length was displayed as a white bar (control cells, *bam>+*) or a gray bar (*Fs(2)RNAi*; *bam>Fs(2)RNAi*). Data are presented as means ± SEMs (*n* > 38 cells). No ProI cells were observed in bam>*Fs(2)KetRNAi* testes. Significance was tested by one-way ANOVA followed by Bonferroni’s post hoc comparison. * *p* < 0.05, ** *p* < 0.01. n.s.: not significant.

**Figure 9 cells-12-02611-f009:**
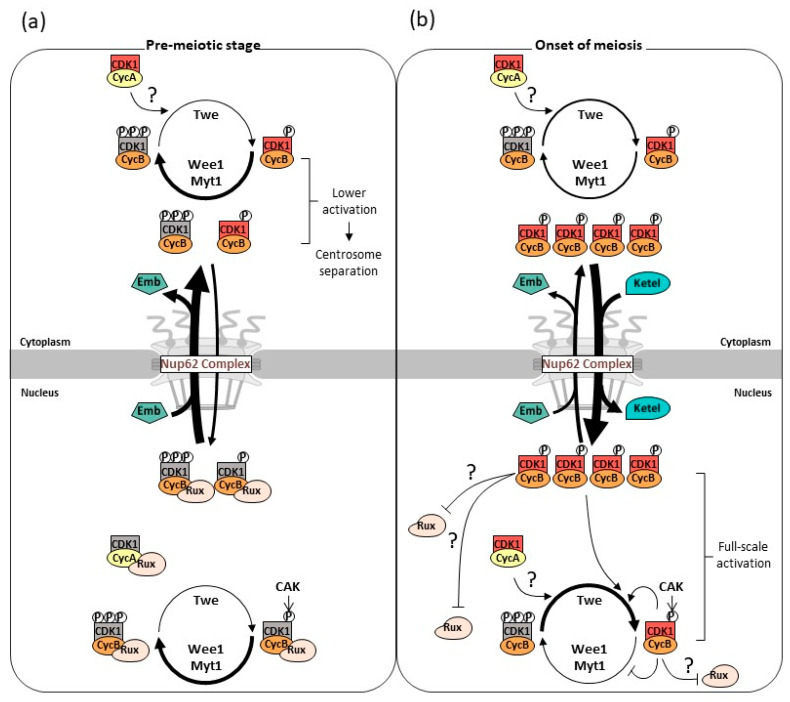
New model for the dynamics of CycB–Cdk1 shuttling in and out of the nucleus during interphase and its rapid nuclear re-entry to initiate male meiosis in *Drosophila*. The model illustrates the dynamics of CycB (orange) and Cdk1 (red) complex transport across the nuclear membrane (light gray) at the pre-meiotic stage when nuclear export via the Nup62 subcomplex in the NPCs continues (**a**). Subsequently, the CycB–Cdk1 complex with the modifications essential for meiotic initiation is transported back into the nucleus to trigger meiotic division (**b**).

## Data Availability

The datasets generated and/or analyzed in the current study are available from the corresponding author on reasonable request.

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
