# Peer review of "Cyclin B Export to the Cytoplasm via the Nup62 Subcomplex and Subsequent Rapid Nuclear Import Are Required for the Initiation of Drosophila Male Meiosis"

_cells, 2023, doi:10.3390/cells12222611_

Round 1
Reviewer 1 Report
Comments and Suggestions for Authors
This interesting research paper by Yamazoe and Inoue examines the localisation of Cyclin B in the specialised- and poorly characterised, male meiotic divisions. It employs the highly tractable model system and combines reverse genetics with imaging to characterise how Cdk1-Cyclin B become active to drive meiotic entry. Overall, the experiments are sound, their logic reasonable and the work increases our understanding of cell division.
However, there are some concerns. First and foremost, the text relies almost entirely on quantitative imaging. The Inoue lab has a strong track record with applying this method to Drosophila. Yet, given the fundamental importance of the technique, it is not described in the methods section. For example, for the PLA analysis Figure 5 K Y-axis is labelled “The number of N/C PLA signal (pixel)” is this a quantification of pixel numbers or signal intensities? All aspects of image quantification across the text need to be outlined to allow reader reproducibility. Some data are over-interpreted, e.g., line 389 “These data confirmed that Cdk was normally phosphorylated by CAK in the nucleus of Nup62-silenced spermatocytes” the data being localisation of CAK and PLA. The data do not directly demonstrate that CAK is phosphorylating Cdk. Rather they are consistent with a model in which CAK activates Cdk in the nucleus of Nup62 depleted cells. Furthermore, line 547 states “Fs(2) Ket-depletion did not influence centrosome separation, which is required for initial activation of Cdk1 before meiosis”. This statement is not accurate. Active Cdk1 is required for centrosome separation but not vice-versa. The text also has typos and possible mis-statements. It needs to be closely proof-read prior to publication. In the version available to me Figure 4 was missing the right-hand portion of two sub-figures. This may be a type-setting error and not amenable by the authors.
Comments on the Quality of English LanguageThe Inoue lab is well-known and highly respected by investigators in the Drosophila and cell cycle communities. Here, the known high standard of the laboratory's work is let down a bit by the English. There are several mistakes and some of the word choices are not entirely appropriate. This has made some statements confusing or questionable. The text would benefit from having a revision where these nuances of the English language were corrected for.
Author Response
1.First and foremost, the text relies almost entirely on quantitative imaging. The Inoue lab has a strong track record with applying this method to Drosophila. Yet, given the fundamental importance of the technique, it is not described in the methods section. For example, for the PLA analysis Figure 5 K Y-axis is labelled “The number of N/C PLA signal (pixel)” is this a quantification of pixel numbers or signal intensities? All aspects of image quantification across the text need to be outlined to allow reader reproducibility.
We appreciate the reviewer for their careful reading and for providing valuable comments. In an attempt to improve our manuscript, we have noted all of the reviewers’ comments and responded to each of the comments one by one.
According to the reviewer’s request, we added a further explanation for the PLA analysis to Materials and Methods (lines 211-212) and the capture in Figure 5k (line 433) as follows: “(k) Ratio of the number of PLA signals in the nucleus to that in the cytoplasm. The number of PLA-positive foci in the nucleus or cytoplasm of each spermatocyte at the S5 stage was counted. The number per pixel in each compartment was calculated. The ratio of the number in the cytoplasm to that in the nucleus (N/C intensity) was displayed on the y-axis.” (lines 433-436). We also revised the capture of (f) as follows; (f) Ratio of the intensity of anti-Cdk7 immunofluorescence in the nucleus to that in the cytoplasm. The fluorescence intensity of the spermatocytes at S5 to Pro was measured. The mean ratio of the intensity in the cytoplasm to that in the nucleus (N/C intensity) was displayed as white bar (control cells) or grey bar (Nup62RNAi cells).” (lines 426-430). We also revised the label on the Y-axis in Figure 5k as "The N/C ratio of the number of PLA foci per pixel.”.
Similarly, we have also revised the manuscript to add an explanation of image quantification in the following Figure legends: Figures 1f (lines 262-266), 1i (lines 272-275), 2a (lines 320-322), 3e (lines 358-363), 4h (397-400), 5f (line 427-430), 6j (line 487-489), 6p (line 493-496), 8b (line 599-602) and 8m (line 612-614).
2.Some data are over-interpreted, e.g., line 389 “These data confirmed that Cdk was normally phosphorylated by CAK in the nucleus of Nup62-silenced spermatocytes” the data being localisation of CAK and PLA. The data do not directly demonstrate that CAK is phosphorylating Cdk. Rather they are consistent with a model in which CAK activates Cdk in the nucleus of Nup62 depleted cells.
We agree with this reviewer’s concern that the co-localization of CAK with Cdk1 and its close association with Cdk1 is not sufficient evidence to conclude that Cdk was normally phosphorylated by CAK. Therefore, according to the suggestion, we revised the sentences in section 4.2 in Discussion as follows: “The data suggesting colocalization and close association of CAK with Cdk1, and those of Twine with Cdk1 in the Nup62RNAi spermatocytes are consistent with a model in which CAK activates Cdk1 in the nucleus of Nup62-depleted cells. Even so, they do not directly prove that Cdk1was phosphorylated and dephosphorylated by the kinase and phosphatase, respectively. Further biochemical experiments are needed to confirm the phosphorylation status of the relevant amino acid residues within Cdk1, although specific antibodies that recognize the phosphorylated peptides in Drosophila are not available. If the speculation above is correct,,,,,” (lines 679-686).
- Furthermore, line 547 states “Fs(2) Ket-depletion did not influence centrosome separation, which is required for initial activation of Cdk1 before meiosis”. This statement is not accurate. Active Cdk1 is required for centrosome separation but not vice-versa.
These cytological phenotypes and cellular localization of CycB are consistent with the interpretation that Fs(2)Ket is required for nuclear import of CycB–Cdk1 to complete full-scale activation of Cdk1. According to the reviewer’s comment, we revised the sentence as follows: “Fs(2)Ket depletion did not influence centrosome separation, which occurred before the full-scale activation of Cdk1 in meiosis (Fig. 8k–m).“ (lines 588-589). As noted by the reviewer, this sentence may mislead the readers. Therefore, we revised the following sentence as follows: “Unlike that in Cdk1- or Nup62-silenced cells (Fig. 4d’–g’), centrosome separation in Fs(2)Ket-silenced spermatocytes was indistinguishable from that of the control cells (Fig. 8i’–l’). Therefore, centrosome separation was not affected by the inhibition of CycB re-entry to the nucleus.” (lines 589-592).
4.The text also has typos and possible mis-statements. It needs to be closely proof-read prior to publication.
We appreciate the reviewer’s careful reading of our manuscript. As the reviewer suggested, we should revise the quality of the English text in this manuscript. We read the manuscript many times ourselves to correct typos in the text and figures and some of the word choices that were inappropriate. We corrected a mis-spelling in Fig. 8a and revised the labels showing genotypes or the notation of gene silencing to remove the spacing to match the text. Furthermore, we also asked two independent professional English proofreaders to improve the quality of the English text including appropriate use of the definite articles. We highlighted altered sentences, phrases, and words using Track Changes in the Word files. We hope that all mistakes and confusing or questionable statements that let down the quality of this study have been improved.
5.In the version available to me Figure 4 was missing the right-hand portion of two sub-figures. This may be a type-setting error and not amenable by the authors.
The missing of the right-hand portion of Fig. 4 was due to our type-setting error in the Word file. We are sorry about the editing mistake. We revised it.
6.Comments on the Quality of English Language
The Inoue lab is well-known and highly respected by investigators in the Drosophila and cell cycle communities. Here, the known high standard of the laboratory's work is let down a bit by the English. There are several mistakes and some of the word choices are not entirely appropriate. This has made some statements confusing or questionable. The text would benefit from having a revision where these nuances of the English language were corrected for.
We appreciate the reviewer’s careful reading of our manuscript. We ourselves reviewed the manuscript many times to find errors, inappropriate expressions, and nuances in the English language. In addition, we asked two independent professional editors who are native English speakers in the Editage Co. to check the quality of the English text and figures. We are going to submit the manuscript revised according to the proofreader’s suggestion together with the quality certificate by the Editage Co. We highlighted all altered sentences, phrases, and words using Track Changes in the Word file.

Reviewer 2 Report
Comments and Suggestions for Authors
In this manuscript, Yamazoe at al find that Cdk1-CycB continuously shuttled into and out of the nucleus in the G2 phase before meiosis. The experimental data suggests that overexpressing WT CycB (but not CycB with nuclear localization signal sequences) can rescue the inhibition of meiosis in Nup62-silenced cells. They found full activation of Cdk1 occurs right after nuclear entry of Cdk1. Cdk1-dependent centriole separation is not observed/affected in Nup62-silenced cells, although the interaction between CDK1 and Cdk-activating kinase/Twine/Cdc25C exists. Evidence suggests roughex silencing can rescue Cdk1 inhibition and initiate meiosis. Thus, nuclear export of Cdk1 may escape the inhibition by cyclin-dependent kinase inhibitor, and importin beta facilitates the reentry of Cdk1 complex to initiate meiosis. It is an interesting study and is suggested to be published in Cancers.
1. Need to improve the figure labeling. For example, it is not easy to find Fig. 1A, 1B, 1C, 1D, 1E, 1G, 1H.
2. In Figure 4. Part of the image was cut off.
Author Response
Reviewer2
Comments and Suggestions for Authors
In this manuscript, Yamazoe at al find that Cdk1-CycB continuously shuttled into and out of the nucleus in the G2 phase before meiosis. The experimental data suggests that overexpressing WT CycB (but not CycB with nuclear localization signal sequences) can rescue the inhibition of meiosis in Nup62-silenced cells. They found full activation of Cdk1 occurs right after nuclear entry of Cdk1. Cdk1-dependent centriole separation is not observed/affected in Nup62-silenced cells, although the interaction between CDK1 and Cdk-activating kinase/Twine/Cdc25C exists. Evidence suggests roughex silencing can rescue Cdk1 inhibition and initiate meiosis. Thus, nuclear export of Cdk1 may escape the inhibition by cyclin-dependent kinase inhibitor, and importin beta facilitates the reentry of Cdk1 complex to initiate meiosis. It is an interesting study and is suggested to be published in Cancers.
- Need to improve the figure labeling. For example, it is not easy to find Fig. 1A, 1B, 1C, 1D, 1E, 1G, 1H.
We appreciate the reviewer for their careful reading and for providing valuable comments. In an attempt to improve our manuscript, we have noted all of the reviewers’ comments and responded to each of the comments one by one.
According to the reviewer’s request, we changed the larger font for labeling. The Journal uses lowercase bold letters to label panels. We have made this change in all figurers and their captions.
- In Figure 4. Part of the image was cut off.
The cut-off of the right-hand portion of Fig. 4 was due to our type-setting error in the Word file. We are sorry about that. We revised it.

Reviewer 3 Report
Comments and Suggestions for Authors
The authors provide extensive data in support of an intriguing and very complex model for the dynamics of CycB-Cdk1 shuttling in and out of the nucleus during interphase and rapid nuclear reentry to initiate male meiosis in Drosophila. Overall, the micrographs are of high quality and data have been quantified and analyzed for statistical significance. The manuscript would be strengthened by addressing the following concerns:
1. Line 223: It should be noted that leptomycin B specifically inhibits CRM1/exportin1. Are there other exportins present in Drosophila that might contribute to CycB export?
2. Line 283-285: Please clarify this statement. The rationale for why reduced cytoplasmic CycB would inhibit meiotic initiation is not clearly stated here.
3. Lines 315-323: It is not clear how the authors are concluding from their data that "full scale activation....spread immediately to the cytoplasm" and that "full scale activation of Cdk1 was suppressed in Nup62-silenced cells." Are the cells shown in Figure 3 Nup62-silenced cells? If so, then this should be clearly stated in the figure legend. If not, then how is this conclusion being drawn?
4. In the PDF manuscript provided by the authors, Figure 4 is cropped on the right and is missing panels from parts F, G, and H. It is not possible to review this incomplete figure or to assess that quality of the data. In addition, the figure caption for Fig. 4 indicates that both Cdk1-silenced and Nup62-silenced cells are shown as "dark grey" columns in the graph. I assume that one should be "light grey" but which is which?
5. Figures 5 and 6: A negative control is need for the proximity ligation assay, otherwise it is not clear if the signals in both the control and Nup62RNAi are not simply false positives. Also, the authors should state more clearly that colocalization alone does not directly prove dephosphorylation activity.
6. Is the slight decrease in N/C shown in Fig. 6P (that has a statistically significant p value) also biologically significant? The interpretation needs to be explained more fully.
Author Response
Reviewer3
Comments and Suggestions for Authors
The authors provide extensive data in support of an intriguing and very complex model for the dynamics of CycB-Cdk1 shuttling in and out of the nucleus during interphase and rapid nuclear reentry to initiate male meiosis in Drosophila. Overall, the micrographs are of high quality and data have been quantified and analyzed for statistical significance. The manuscript would be strengthened by addressing the following concerns:
- Line 223: It should be noted that leptomycin B specifically inhibits CRM1/exportin1. Are there other exportins present in Drosophila that might contribute to CycB export?
We appreciate the reviewer for their careful reading and for providing valuable comments. In an attempt to improve our manuscript, we have noted all of the reviewers’ comments and responded to each of the comments one by one.
According to the reviewer’s comment, we revised the phrase “,,,, in the presence of LMB, which inhibited CRM1/exportin1 specifically,” on lines 246-247 in the current manuscript, and stated as requested “leptomycin B (LMB) that specifically inhibits CRM1/exportin1”. In Drosophila, only one Exportin orthologue, Emb has been identified. The leptomycin B is considered to act on and inhibit this unique orthologue of CRM1/exportin.
- Line 283-285: Please clarify this statement. The rationale for why reduced cytoplasmic CycB would inhibit meiotic initiation is not clearly stated here.
We agree that the explanation described in previous lines 283-285 was insufficient as a rationale for reaching the conclusion that the reduced cytoplasmic CycB would inhibit meiotic initiation. We revised this sentence and added one underlined sentence: (line 304) “Interestingly, the overexpression of CycB in Nup62-silenced cells resulted in the generation of post-meiotic cysts consisting of 32 cells (12.5% of total spermatid cysts examined, 4 out of 32) and 64 cells (56.2% of the spermatid cysts, 18 out of 32), whereas ectopic overexpression of NLS-CycB produced cysts containing 16 spermatids only (n = 24) (Fig. 2b). These genetic data that inhibition of meiotic initiation in Nup62-silenced spermatocytes was rescued by overexpression of CycB, but not NLS-CycB, suggested that CycB supply to the cytoplasm was more effective than that to the nucleus for the rescue from the cell cycle arrest. In other words, these data suggest that reduced cytoplasmic CycB, rather than the accumulation of ectopic nuclear CycB, was involved in the inhibition of meiotic initiation.” (lines 304-314).
- Lines 315-323: It is not clear how the authors are concluding from their data that "full scale activation....spread immediately to the cytoplasm" and that "full scale activation of Cdk1 was suppressed in Nup62-silenced cells." Are the cells shown in Figure 3 Nup62-silenced cells? If so, then this should be clearly stated in the figure legend. If not, then how is this conclusion being drawn?
The anti-MPM2 immunostaining presented in Fig. 3a-d was performed against normal spermatocytes at S5 to prophase I stage. The results of anti-MPM2 immunostaining of Nup62RNAi cells had already been published in Fig. 2e and f in our previous paper (Okazaki et al., 2020). Although we also confirmed this point in this study, we avoided publishing the same results again. As pointed out by the reviewer, the final sentence was inadequate in the Results as there was a logical gap. Considering the reviewer’s concern, we revised the relevant sentences as follows: (Before the sentence on line 347, we added the underlined sentence.) (line 345)“This sharp increase in the MPM2 epitopes suggested that the full-scale activation of Cdk1-CycB began and flourished in this period. These observations indicated that full-scale activation of Cdk1-CycB occurred at the nucleus after its rapid reentry, and it spread immediately to the cytoplasm from earlier ProI to PrometaI (Fig. 3e), when Sa-GFP foci were no longer detected (Fig. 3d”’). The robust intensity of anti-MPM2 signal in Nup62-silenced cells was not observed in the previous report (Okazaki et al., 2020)” (line 351). And, we removed the final sentence (it was on the previous line 323) from the current manuscript.
- In the PDF manuscript provided by the authors, Figure 4 is cropped on the right and is missing panels from parts F, G, and H. It is not possible to review this incomplete figure or to assess that quality of the data.
The missing of the right-hand portion of Fig. 4 was due to our type-setting error in the Word file. We are sorry about the editing mistake. We corrected this error.
In addition, the figure caption for Fig. 4 indicates that both Cdk1-silenced and Nup62-silenced cells are shown as "dark grey" columns in the graph. I assume that one should be "light grey" but which is which?
Data for Nup62-silenced cells should be represented in light grey columns. As requested by the reviewer, we revised the mistake and showed bars in color; “The length between paired centrosomes were measured in each spermatocyte and the mean length was displayed as a white bar (control cells, bam>+), yellow bar (Cdk1RNAi; bam>Cdk1RNAi), and red bar (Nup62RNAi; bam>Nup62RNAi).” (lines 397-400).
- 5. Figures 5 and 6: A negative control is need for the proximity ligation assay, otherwise it is not clear if the signals in both the control and Nup62RNAi are not simply false positives.
According to the reviewer’s comment, we added the data of the PLA using anti-Cdk1 antibody alone in Fig. S2a, b and those using anti-Cdk7 antibody alone in Fig. S2c, d as negative controls for the experiments in Fig. 5g, h. As neither antibody alone provided enough positive signals of the PLA, we believed the PLA signals obtained using both antibodies were real positives. Similarly, we presented the PLA data using anti-GFP antibody alone in Fig. S2e, f. These are the negative controls for the PLA shown in Fig. 6k-o. These data also provided pieces of evidence that the PLA signals using both antibodies in Fig. 5 and 6 were real positives. Within this short 10-day given for revision, we could not perform similar control experiments using the RNAi cells, which require more than two weeks.
- Also, the authors should state more clearly that colocalization alone does not directly prove dephosphorylation activity.
As we agree this reviewer’s concern, we added the following statement in the Discussion:(line 679) “The data suggesting colocalization and close association of CAK with Cdk1, and those of Twine with Cdk1 in the Nup62RNAi spermatocytes are consistent with a model in which CAK activates Cdk1 in the nucleus of Nup62-depleted cells. Even so, they do not directly prove that Cdk1was phosphorylated and dephosphorylated by the kinase and phosphatase, respectively. Further biochemical experiments are needed to confirm the phosphorylation status of the relevant amino acid residues within Cdk1, although specific antibodies that recognize the phosphorylated peptides in Drosophila are not available.” (line 686).
- Is the slight decrease in N/C shown in Fig. 6P (that has a statistically significant p value) also biologically significant? The interpretation needs to be explained more fully.
As we presented in Fig. 6n and o, the PLA signals are distributed in the nucleus and the cytoplasm in both control cells and Nup62RNAi cells. The N/C ratios of the signal a bit exceeded 1. 0 (Fig. 6p). The average ratio of the signal was only slightly lower than that in Nup62RNAi cells. Although the differences were statistically significant, we do not know whether it has any impact on the biological outcome. Rather, we stated that the PLA signals can be seen in whole regions of the spermatocytes in control and Nup62RNAi cells. Therefore, we added the following sentence on line 457: “The PLA signals can be seen in both the nucleus and the cytoplasm of the spermatocytes in control and Nup62RNAi cells to the extent that the N/C ratio slightly exceeded 1.0 (Fig. 6p).” (line 459).

Round 2
Reviewer 3 Report
Comments and Suggestions for Authors
The authors have addressed all my comments.